# Comparison between Diffusion-Weighted Sequences with Selective and Non-Selective Fat Suppression in the Evaluation of Crohn’s Disease Activity: Are They Equally Useful?

**DOI:** 10.3390/diagnostics10060347

**Published:** 2020-05-27

**Authors:** Ilze Apine, Reinis Pitura, Ivanda Franckevica, Juris Pokrotnieks, Gaida Krumina

**Affiliations:** 1Children Clinical University Hospital of Riga, LV 1004 Riga, Latvia; ivanda.franckevica@bkus.lv; 2Department of Radiology, Riga Stradin’s University, LV 1004 Riga, Latvia; reinis.pitura@gmail.com (R.P.); gaida.krumina@rsu.lv (G.K.); 3Department of Pathology, Riga Stradin’s University, LV 1007 Riga, Latvia; 4Department of Internal Diseases, Riga Stradin’s University, LV 1007 Riga, Latvia; pokrot@latnet.lv

**Keywords:** MR enterography, terminal ileitis, diffusion-weighted imaging, DWI, DWIBS, ADC, MaRIA, Clermont score, DWI fat suppression techniques

## Abstract

*Background*: We compared the efficiency of two MRI diffusion weighted imaging (DWI) techniques: DWI with SPIR (DWI_SPIR_) and DWI with STIR (DWI_STIR_), to estimate their eligibility for quantitative assessment of Crohn’s disease activity in children and adults. *Methods*: In inflamed terminal ileum segments (*n* = 32 in adults, *n* = 46 in children), Magnetic Resonance Index of Activity (MaRIA) was calculated, ADC values of both DWI techniques were measured, and the corresponding Clermont scores calculated. ADC values of both DWI techniques were compared between both and within each patient group, assessing their mutual correlation. Correlations between MaRIA and the corresponding ADC values, and Clermont scores based on both DWI techniques were estimated. *Results*: No correlation between ADC of DWI_SPIR_ and DWI_STIR_ was observed (rho = 0.27, *p* = 0.13 in adults, rho = 0.20, *p* = 0.17 in children). The correlation between MaRIA and Clermont scores was strong in both techniques—in SPIR, rho = 0.93; *p* < 0.0005 in adults, rho = 0.98, *p* < 0.0005 in children, and, in STIR, rho = 0.89; *p* < 0.0005 in adults, rho = 0.95, *p* < 0.0005 in children. The correlation between ADC and MaRIA was moderate negative for DWI_STIR_ (rho = 0.93, *p* < 0.0005 in adults, rho = 0.95, *p* < 0.0005 in children), but, in DWI_STIR_, no correlation between ADC and MaRIA score was observed in adults (rho = −0.001, *p* = 0.99), whereas children presented low negative correlation (rho = −0.374, *p* = 0.01). *Conclusions*: DWI_STIR_ is not suitable for quantitative assessment of Crohn’s disease activity both in children and adult patients.

## 1. Introduction

Crohn’s disease (CD) is an idiopathic chronic relapsing inflammatory bowel condition, which manifests itself in fistulae, abscesses and strictures [1]. The chronic inflammation causes progression of intestinal complications, requiring surgery in most cases [2,3]. It is known that active therapy including adjustment of medication reduces the number of complications and the need for surgery [1]. The goal of treatment for CD is either resolution of abdominal complaints and endoscopically confirmed remission, or resolution of inflammatory signs in cross-sectional imaging [4]. Since 2019, according to ECCO-ESGAR Guideline for Diagnostic Assessment in inflammatory bowel diseases, cross-sectional imaging modalities are recognised to be an alternative for the evaluation of the disease activity [5]. Magnetic resonance enterography (MRE) is proven to have a great potential for evaluating CD activity, due to its high soft tissue resolution, non-invasiveness and lack of ionizing radiation, and ability to obtain findings not only within but also around the bowel wall [6].

To assess CD activity, a number of MRI scoring systems have been developed for standardising the measured outcomes [6]. Among these indices, Magnetic Resonance Index of Activity (MaRIA) is the only approved MRE-based score, having the strongest validation data based on a large population of patients [7]. However, MaRIA score relies on the measurement of enhanced gadolinium contrast media known to be related to systemic nephrogenic fibrosis [8], as well as accumulation of gadolinium deposits in the brain [9] and other body tissues, such as skin, liver and bones [10]. As patients with CD usually require multiple follow-up examinations which expose them to a certain risk of gadolinium accumulation, solutions allowing replacement of the contrast medium administration are of importance.

Diffusion-weighted imaging (DWI) has been shown to have potential as a replacement of the contrast media, by detecting lesions prior their appearance in conventional images [11]. It also outperforms the T1 post-contrast dynamic series [12,13,14,15]. DWI is proven to be useful for the detection of inflammation in CD [16,17], and, when added to the MRE protocol, it yields increased sensitivity in diagnostics of CD than conventional MRE alone [18] thus potentially supporting early diagnosis of CD. DWI also has a very important role in evaluating CD activity, which allows assessment of treatment efficacy [7].

DWI can be measured quantitatively with apparent diffusion coefficient (ADC). ADC of DWI is proven to be as useful as the relative contrast enhancement (RCE) used for calculation of MaRIA [19]. A DWI-based MaRIA, or Clermont score, is designed to serve as an alternative to avoid gadolinium administration [17] and is reported to have excellent correlation compared to MaRIA [20]. This index still is to be validated by confirmatory studies.

Due to the low sensitivity to motion-induced phase errors and advantages of short imaging times, DWI normally uses single-shot echo-planar imaging (EPI) acquisition, related to the presence of susceptibility artefacts at tissue interfaces [21] and chemical shift-induced ghosting artefacts deflecting the fat signal several pixels away from the water signal. To avoid these artefacts, DWI protocols should include fat saturation techniques [22,23]. These techniques are simply divided into two categories—fat, or spectral, selective and non-selective ones.

Among fat selective techniques, chemical shift selective fat suppression (CHESS) is a common fat saturation technique in which an excitation pulse with a bandwidth selective to the resonance frequency of fat is applied, followed by a spoiler dephasing gradient [24]. There are also two fat selective hybrid sequences. In spectral attenuated inversion recovery (SPAIR) the fat signal is inverted with an adiabatic spectrally selective pulse, and acquisition starts from the moment of inversion time (TI) that nulls the fat signal. Spectral pre-saturation with inversion recovery (SPIR), in turn, is a hybrid sequence with a spectrally selective inversion pulse is applied tuned to fat frequency [24]. After the TI, a conventional 90° spin-echo pulse is applied to saturate just the fat signal [25]. All these techniques use spectrally selective pulses suppressing signal from only fat protons.

Alongside fat selective saturation techniques, the DWI weighted sequence with Short Tau Inversion Recovery (STIR) fat suppression technique is also used. STIR is an inversion recovery technique based on the difference in relaxation between water and fat tissues, which have a much shorter T1 relaxation time compared to other tissue types. An inversion pulse is applied before the excitation pulse; the spins of all tissues invert and then perform T1 relaxation. By selecting the inversion time (TI) such as the longitudinal magnetisation of fat at that time is zero, fat spins will not participate to the MR signal [26]. STIR-based DWI had been developed in 2004 by the research group of Takahara who named it Diffusion Weighted Imaging with Background Body Signal Suppression (DWIBS). This sequence allows free breathing, permitting multiple slice excitation and signal averaging over an extended period to average motion [27]. When compared to the selective fat saturation techniques, while chemical fat selective saturation techniques are more influenced by magnetic field inhomogeneities [28] the use of DWI with non-selective STIR allows robust and more homogenic fat suppression within large body regions [27,28] including off-center localisations [24], providing a higher contrast-to-noise ratio (CNR) [23], insensitivity to magnetic field inhomogeneities, and decreased image distortion [29] making it suitable for scanning large areas [23]. It also suppresses the signals from bowel content [29]. DWIBS was initially developed for the whole-body imaging of oncology patients to detect tumour relapse and metastases [27], but it is now used in many other applications including detecting inflammatory lesions, abscesses, intravascular thrombi, and visualisation of the peripheral nerves [23]. DWI with STIR technique is reported to have better image quality and less image artefacts when compared to conventional, or spectral selective, DWI [30]. Regarding ADC analysis, DWI with STIR is proven to be superior over conventional DWI, in the assessment of breast lesions [31]. The American College of Radiologists recommends including DWI with STIR in standard MRE protocols for imaging of the gastrointestinal tract [32]. Using DWI sequence with STIR in MRE examinations is further recommended by a number of authors [33,34]. It is also reported to be used in the assessment of CD activity [35]. The non-selectivity of STIR can be a disadvantage if tissues contain other substances with short T1 time, such as methaemoglobin, mucoid tissue, proteinaceous material, and melanin [24]. Another disadvantage of STIR-based fat suppression, is decreased SNR by partial loss of proton signal during the inversion time [29], causing grainy image appearance. Nevertheless, the free breathing DWIBS as DWI sequence with STIR is one of the most important DWI-related discoveries, allowing whole body imaging [27], and so improving diagnosis in oncology (tumour staging, detection of tumour relapse, monitoring response to therapy) [36]. Despite the reduced SNR, DWIBS contributes in improved detection of subtle lesions due to higher CNR [23]. DWI with STIR is reported to be feasible for the identification and characterisation of lymph nodes in patients with uterine cervical cancer [37]. ADC calculated from DWI_STIR_ (ADC-DWI_STIR_) tracking images is reported to be superior over ADC of DWI_SPIR_ (ADC-DWI_SPIR_) in differentiating between malignant and benign breast lesions [31] whilst also being non-dependent on motions [38], important in bowel imaging. However, there have been no studies published on comparison between DWI sequences with non-selective STIR and chemical fat selective DWI techniques for quantitative assessment of CD activity.

The aim of our study was to compare the performance of DWI sequences with fat selective Spectral Presaturation with Inversion Recovery (SPIR) (DWI_SPIR_) and non-selective STIR, or DWIBS, (DWI_STIR_), in quantitative assessment of active CD in the terminal ileum through the following measures:(1)Measuring ADC values of DWI with SPIR (ADC-DWI_SPIR_) and DWI with STIR (ADC-DWI_STIR_) in groups of adult and paediatric patients, comparing each individual patient group and assessing their mutual correlation,(2)comparing ADC- DWI_SPIR_ and ADC-DWI_STIR_ values between the groups of adults and pediatric patients,(3)estimating correlations of ADC-DWI_SPIR_ and ADC-DWI_STIR_ values with the corresponding MaRIA, calculated from the contrast–enhanced sequences within the same bowel segments,(4)calculating Clermont scores values based on ADC-DWI_SPIR_ and ADC-DWI_STIR_ and estimating their correlation with MaRIA within the same bowel segments.

## 2. Materials and Methods

### 2.1. Patient Population

In this prospective observational cross-sectional study, the patients underwent MRE examination between April 2016 and April 2019. All patients involved in the research had either symptomatic CD, or underwent MRE examination for monitoring the disease course under treatment. The faecal calprotectin levels in all study subjects exceeded 1000 μg/g. The inclusion criteria were: proven active non-stricturing non-penetrating CD in the terminal ileum, presenting with thickened bowel wall (thickness > 3 mm), presence of mural oedema (hyperintensity of the bowel wall in T2-weighted images compared to the psoas muscle) [39], signs of restricted diffusion in both of DWI sequences with SPIR and STIR presenting with high SI in DWI tracking images of b = 800 s/mm^2^ along with low signal intensity (SI) in the ADC map, and early mucosal hyperenhancement in the post-Gd series [40]. The exclusion criteria were: locations of CD other than the terminal ileum, bowel thickness less than 3 mm, dynamic blurring in either of the DWI or T1 post-Gd images, inability to locate active bowel wall inflammation in both DWI sequences, with SPIR and STIR, and post-Gd T1 within one and the same segments. The study was performed in accordance with the Declaration of Helsinki and approved by the Ethics committee of Riga Stradin’s University on 10 September 2015, its permission number was 6/10.09.2015.

### 2.2. MRI Technique

All patients fasted for 6 h prior the MRE examination, being allowed to intake only water. No bowel cleansing was carried out. The bowel distension was maintained with 1.000–1.500 mL of 2.5% mannitol solution, consumed slowly before the MRE procedure for 45 min. After that, patients were asked to lie in the right decubitus position, and they received another 250 mL of 2.5% mannitol solution to intake slowly for another 20 min. The MRE examinations were performed with a 1.5T scanner (Ingenia, Philips Medical Systems, Best, The Netherlands) covering the region from the diaphragm to the pelvis with a 16-element phased array body coil. The patients were scanned in the prone position. The MRE protocol included:(1)coronal bTFE (Balanced Turbo Field Echo) cine sequence for real-time assessment of the bowel peristalsis,(2)axial DWI sequence with SPIR, using diffusion factors b fixed at 0, 600 and 800 s/mm^2^, with corresponding ADC maps,(3)axial DWIBS sequence (DWI with STIR), using diffusion factors b fixed at 0, 600 and 800 s/mm^2^, with corresponding ADC maps,(4)axial and coronal T2-weighted sequences without fat suppression (T2 TSE),(5)axial and coronal T2-weighted sequences with fat suppression (T2 SPAIR),(6)coronal T2 fat suppression magnetic resonance cholangiopancreatography (MRCP) sequence with radial 3D reconstructions,(7)coronal T1-weighted dynamic postcontrast images e-THRIVE (T1 high-resolution isotropic volume excitation), followed by delayed post-contrast axial e-THRIVE images.

The scanning parameters of the DWI_SPIR_ and DWI_STIR_ (DWIBS) protocols are given in Table 1. To reduce bowel peristalsis, hyoscine butylbromide (Buscopan, Sanofi, Athens, Greece) was intravenously administered, prior to the DWI_SPIR_ and DWI_STIR_ sequences and the coronal dynamic contrast sequences. The dosage was 10 mg in patients under 50 kg and 20 mg in patients 50 kg or above, diluted in 20 mL of saline solution.

### 2.3. MR Image Analysis

The altered locations of the terminal ileum were identified and divided into approximately 3 cm long segments. The total number of segments was 78, 32 in adults and 46 in paediatric patients. In each segment, wall thickness was measured in mm, presence of ulcers (present or absent) was estimated, and six measurements of ADC-DWI_SPIR_ and ADC-DWI_STIR_ in the corresponding DWI_SPIR_ and DWI_STIR_ tracking images of b = 800 s/mm^2^ were performed in each segment, in the zone of the highest signal intensity within the bowel wall. Six measurements of the wall signal intensity were taken in the same location both before (WSI-preGd) and after (WSI-postGd) administration of gadolinium contrast medium, in the site with the highest SI in the postcontrast images. Six measurements of standard deviation (SD) representing the image noise were performed outside the body before (SD-preGd) and after (SD-postGd) administration of gadolinium contrast medium [39]. The mean value of all measurements was used for calculations. In each altered bowel segment, MaRIA was calculated using the following formula:MaRIA = 1.5 × wall thickness (mm) + 0.02 × RCE + 5 × oedema + 10 × ulcers,
where the presence or absence of ulcers and oedema was rated as 1 or 0, accordingly. RCE was calculated as follows:RCE = (WSI-postGd − WSI-preGd)/(WSI-preGd)) × 100 × (SD-preGd/SD-postGd),
where the SD-preGd and SD-postGd corresponded to the mean of the six SD values of SI, measured outside of the body before and after gadolinium administration, accordingly [39]. Since oedema was one of the inclusion criteria representing inflammation, it was always present, and its rating was always equal to 1. The Clermont score, or DWI-MaRIA, for both DWI_SPIR_ and DWI_STIR_ sequences, was calculated per formula [20]:DWI-MaRIA = 1.646 × bowel thickness − 1.321 × ADC + 5.613 × oedema + 8.306 × ulceration + 5.039

The i/v gadolinium contrast agent used before October 2018 for all adult patients and all but two children was gadodiamide (Omniscan 0.05 mmol/mL, GE Healthcare, Cork, Ireland, dosage 0.2 mL/kg, or 0.1 mmol/kg). Gadobutrol (Gadovist 1 mmol/mL, Bayer, Berlin, Germany, dosage 0.1 mL/kg, or 0.1 mmol/kg) was used for the two paediatric patients examined after October 2018.

The ADC, WSI and SD measurements were performed using 4–9 mm^2^ oval region of interest (ROI). The image analysis and the measurements were performed by one radiologist with 19 years’ experience in abdominal MRI imaging. The review of images and ADC measurements were performed using a dedicated Philips Intellispace Portal postprocessing server, v. 5.0 (Philips Medical Systems, Best, The Netherlands, 2014). The WSI and image noise measurements were performed using Clear Canvas DICOM Viewer, v. 13.2 (Synaptive Medical, Toronto, ON, Canada, 2019).

### 2.4. Statistical Analysis

The statistical analysis was performed using software SPSS 20.0 (IBM Corporation, Armonk, NY, USA, 2011). The median values with standard deviations for ADC-DWI_SPIR_ and ADC-DWI_STIR_, MaRIA, DWI_SPIR_-Clermont, and DWI_STIR_-Clermont scores were calculated. 95% CI was calculated for median differences. The statistical significance of differences between the groups was determined using Wilcoxon signed rank test. Spearman’s correlation coefficient was used to assess the correlations between quantitative parameters. *p* values of <0.05 (two-tailed) were chosen as a level of statistical significance. The Bonferroni correction was used to control Type 1 error in multiple comparisons.

## 3. Results

Seventeen patients: five adults (23, 25, 36, 40 and 57 years old) and 12 children (11 years old; *n* = 2, 12 years old; *n* = 3, 13 years old; *n* = 1, 14 years old; *n* = 4, 17 years old; *n* = 2) were enrolled in the study. In one adult patient, the duration of medical history prior to the MRE examination was more than two years in three adult patients—from 6 till 12 months, but in one adult patient, CD was asymptomatic, of unknown length, and it was detected upon performing a set of infertility tests. In one paediatric patient, the duration of CD was slightly less than two years, but in the remaining 11 patients the duration of medical history was less than 6 months.

The overview of measured ADC-DWI_SPIR_ and ADC-DWI_STIR_ values, as well as calculated values of MaRIA, DWI_SPIR_-based Clermont score and DWI_STIR_-based Clermont score, is presented in Table 2.

There was a statistically significant difference of 10.32% (*p* = 0.02) between the median values of ADC-DWI_SPIR_ in adults and children, appearing lower in children than in adults; no statistically significant difference (*p* = 0.38) between ADC-DWI_STIR_ in adults and children was detected. There was statistically significant difference of 8% (*p* = 0.03) between ADC-DWI_SPIR_ and ADC-DWI_STIR_ values in adults, appearing lower in DWI_STIR_, but no statistically significant difference (*p* = 0.97) between ADC-DWI_SPIR_ and ADC-DWI_STIR_ values in children. The graphical comparative distribution of the ADC-DWI_SPIR_ and ADC-DWI_STIR_ values between both patient groups is shown in Figure 1a,b. The graphical comparative distribution of the ADC-DWI_SPIR_ and ADC-DWI_STIR_ values within each of the patient groups is shown in Figure 2a,b.

In all patients of both groups, the MaRIA value corresponded to active disease (i.e., ≥7) [6]. Excluding one patient with the score value of 10.65, MaRIA score values in all adult patients also corresponded to severe disease (i.e., ≥11). In the paediatric group, in all but three patients, with the values of 9.95, 10.25 and 10.66, MaRIA values exceeded 11 thus corresponding to severe disease [6].

In all patients of both groups, the DWI_SPIR_-based Clermont score value corresponded not only to active disease (i.e., >8.4) but severe disease (i.e., ≥12.5). In two adult patients, the DWI_STIR_-based Clermont score values (i.e., 5.92 and 8.20) were below the threshold of 8.4 for active disease, however the values of all other patients corresponded both to active and to severe disease. In one paediatric patient, the DWI_STIR_-based Clermont score value (i.e., 8.24) was slightly below the threshold of 8.4 for active disease; two patients with values of 11.26 and 12.38 corresponded to active disease, and all other patients corresponded to severe disease [6].

The correlation between ADC-DWI_SPIR_ and ADC-DWI_STIR_ was weak and statistically unreliable in both adults (rho = 0.27; *p* = 0.13) (Figure 3a) and children (rho = 0.22; *p* = 0.15) (Figure 3b).

There was a strong and statistically significant correlation between MaRIA and ADC-DWI_SPIR_-based Clermont score in both adults (rho = 0.93; *p* < 0.0001) (Figure 4a) and in children (rho = 0.98; *p* < 0.0001) (Figure 4b). There was also a strong and statistically significant correlation between MaRIA and ADC-DWI_STIR_-based Clermont score in adults (r = 0.89; *p* < 0.0001) (Figure 5a) and in children (rho = 0.95; *p* < 0.0001) (Figure 5b). The correlation between ADC-DWI_SPIR_ and MaRIA was moderate negative and statistically reliable in both adults (r = −0.50, *p* = 0.004) (Figure 6a) and children (r = −0.54, *p* < 0.0001) (Figure 6b). There was no correlation between ADC-DWI_STIR_ and MaRIA (rho = −0.001, *p* = 0.99) in adults (Figure 7a), and low negative statistically reliable correlation (rho = −0.374, *p* = 0.01) in children (Figure 7b).

## 4. Discussion

The primary goal of treatment for CD is to achieve remission. Although the primary endpoint of treatment has long been endoscopic remission i.e., mucosal healing [5,39,40], CD is a transmural inflammation. Even in patients with sustained mucosal healing, transmural inflammation may persist [41,42,43]. It is proven that compared to mucosal healing, transmural healing is related to improved long-term outcomes, including sustained long-term steroid-free clinical remission, less need for rescue therapy, less CD-related hospitalisations and CD-related surgery [44]. Therefore, transmural healing has recently been proposed as a new target for CD treatment [45]. To assess transmural changes, imaging techniques are required allowing evaluation of the altered intestinal wall along its entire length and thickness. MRE, being an informative non-invasive radiation-free cross-sectional imaging modality providing high soft tissue resolution, is proven to have a potential to replace endoscopy in assessment of inflammatory activity [39].

MaRIA score, developed by the Rimola research team [39], is the only validated index for measuring inflammatory activity in the ileum distal loop, tested in large patient populations [7] and multicenter research [46]. However, multiple gadolinium contrast injections should be avoided, due to related adverse effects [9,10,11]. In 2019, this same research group also published a report on the simplified MaRIA (sMaRIA), based on data from 98 patients, accounting fat stranding, instead of RCE [46]. However, this novelty approach requires larger validation studies. The London index measures the wall thickness and presence of edema [47], providing an accurate assessment of inflammatory activity, however cannot be used in estimating disease severity, and thus is not applicable in the evaluation of therapeutic response upon follow-up examinations [46]. The Clermont score is based on DWI (therefore called DWI-MaRIA), and it was derived by the research group of Clermont-Ferrand university as an alternative for MaRIA, replacing RCE with ADC, thus avoiding administration of gadolinium contrast media. The authors of the Clermont score state it is not only useful in estimation of ileal CD activity, with excellent correlation with RCE-based MaRIA [20], but also in the detection of ulcers [48] and prediction of remission after biological therapy [49]. However, the performance of the Clermont score still should be validated.

In DWI, which the Clermont score is based on, tissue contrast relies on differences of motions of water molecules among various tissues [50]. In each MRI system, there are several choices of fat saturation techniques for use with DWI. At our institution, the MR protocol repository for abdominal imaging contains two types of DWI sequences—DWI with SPIR fat suppression technique and DWIBS sequence using STIR; their scanning parameters are given in Table 1. When compared to the DWI with SPIR, bowel walls in DWIBS visually look sharper and contours of structures are better delineated (Figure 8). Precise delineation of inflammatory lesions could be very important in diagnosis of CD, which can be subject to delay for up to several years [51], especially in locations not assessible by endoscope, where cross-sectional imaging could be the only solution to reveal inflammation. Therefore, due to better delineation of bowel walls in DWI with STIR, our goal was to assess the reliability of ADC-DWI_STIR_ measurements to be used in calculating the Clermont score, when compared to the ADC-DWI_SPIR_ values.

In the current literature, there are no strictly defined recommendations regarding the best fat suppression techniques to be used in DWI for evaluation of CD activity. Singha et al. and Park recommend using DWI sequence with STIR within protocols for assessment of gastrointestinal tract mentioning the role of ADC measurement [33,34]. The American College of Radiologists recommends that DWI sequence with STIR should be included within the MRE protocol [32]. The free-breathing technique affords multiple averages, thus leading to better SNR, so probably having advantages in performing measurements of ADC [29]. To our knowledge of using ADC in the assessment of bowel inflammation, Kiryu et al. and Caruso et al. are the only research teams reporting DWI with STIR used in ADC measurements for quantitative assessment of bowel information [35,52]. The Caruso team also used this ADC exactly to calculate the Clermont score. However, several other authors emphasise using free breathing techniques [13,14,18,19,20,53,54,55], without specifying the fat saturation method used with DWI. Therefore, there is a chance that researchers consider navigator triggered DWI sequences with SPIR, SPAIR, and CHESS requiring less signal averaging than free breathing techniques, however, in some of these studies, the related fat saturation technique could theoretically be STIR. Consequently, if the article does not explicitly state the technique of fat saturation, it is impossible to judge exactly which one is used.

Initially, we calculated the correlation between ADC-DWI_SPIR_ and ADC-DWI_STIR_ with the expectation of a good correlation. To our surprise, although ADC-DWI_SPIR_ and ADC-DWI_STIR_ visually seemed to be comparable, we observed almost no correlation between ADC-DWI_SPIR_ and ADC-DWI_STIR_ measured in one and the same bowel segments in both adults (rho = 0.27; *p* = 0.13) and children (rho = 0.22; *p* = 0.15). Although DWI_STIR_ is performed under free breathing, and availability of both repeated stimulations and acquisitions contributes to improved SNR and both spatial and temporal resolution [30] the allowance of respiratory motion in DWI_STIR_, means that slice levels of images obtained with different b-values may not be identical. Since DWIBS employs multiple slice excitations, slice levels of images obtained with the same b-value may be different [29]. The weak correlation between ADC-DWI_SPIR_ and ADC-DWI_STIR_ values may also be impacted by the conceptually different fat suppression mechanisms of SPIR and STIR, on ADC values of the intestinal wall in relation to histopathological characteristics of bowel inflammation, due to differences in gut wall histopathology in adults and children. This issue will be discussed further in the article.

Within the study, the ADC-DWI_SPIR_ and ADC-DWI_STIR_ values were analysed in two dimensions:(1)ADC values of both adults and children were compared within a single fat suppression technique, and we observed statistically significant ADC-DWI_SPIR_ difference between adults and children (1.31 × 10^−3^ mm^2^/s, SD 0.29, vs. 1.16 × 10^−3^ mm^2^/s, SD 0.31; *p* = 0.02), with 12.12% lower ADC values in children compared to adults, but no statistically significant difference between the ADC-DWI_STIR_ values in adults and children (1.09 × 10^−3^ mm^2^/s, SD 0.49, vs. 1.20 mm^2^/s × 10^−3^, SD 0.44; *p* = 0.38);(2)both DWI_SPIR_ and DWI_STIR_ techniques were compared within one patient group, both in adults and children. In this case, the analysis showed difference of 16.73% between ADC-DWI_SPIR_ and ADC-DWI_STIR_ values in adults, being lower in DWI_STIR_ (1.31 × 10^−3^ mm^2^/s, SD 0.29, vs. 1.09 × 10^−3^ mm^2^/s, SD 0.49; *p* = 0.03), but did not show difference between ADC-DWI_SPIR_ and ADC-DWI_STIR_ values in the children’s group (1.16 mm^2^/s × 10^−3^, SD 0.31, vs. 1.20 × 10^−3^ mm^2^/s, SD 0.44, *p* = 0.97).

These observations raise questions about why ADC-DWI_SPIR_ values are lower in paediatric patients than in adults whilst no difference in ADC-DWI_STIR_ between adult and paediatric patients is observed, and why the ADC-DWI_STIR_ values are lower than ADC-DWI_SPIR_ in adults, whilst there is no difference between ADC-DWI_SPIR_ and ADC-DWI_STIR_ values in pediatric patients. In answering these questions, either a differing histopathological pattern in adult and paediatric CD, or differences between the fat suppression mechanism of DWI_SPIR_ and DWI_STIR_ techniques influences ADC values, or combination of both factors must be taken into account.

To interpret our results, it is necessary to consider the pathophysiological characteristics of the tissue, the physical basis of both DWI sequences and the duration of CD history. Although the exact cause of the restricted diffusion in CD still remains unknown, the three ruling theories considered are as follows: (1) narrowing of extracellular space in active CD (caused by presence of oedema and increased cell density due to migration of inflammatory cells, mostly lymphocytes), formation of lymphoid aggregates into the lamina propria and submucosa of the inflamed wall segments (Figure 9a), presence of dilated lymphatic vessels and epithelioid granulomas, and formation of micro-abscesses) [56,57,58,59], (2) increased perfusion, and (3) mural fibrosis [56,57]. Although the morphological pattern of CD is generally similar in adult and pediatric patients [59], the main difference between the histopathology of pediatric and adult CD, is the more frequent appearance of epithelioid granulomas in the inflamed bowel wall of children [58,60,61] (Figure 9b).

Whenever analysing the performance of ADC-DWI_STIR_, the non-selectivity of STIR fat suppression must be considered. Unlike the SPIR technique which uses spectral selective radiofrequency pulse suppressing solely the fat signal, STIR technique uses an inversion recovery technique based on the T1 relaxation time of the tissues examined. Apart from fat having a short T1 time, STIR technique suppresses other substances of short T1 time, thus adding to decrease of ADC value by suppression of signal from methaemoglobin, melanin, mucoid tissue, and proteinaceous fluid [24]. When considering the presence of methaemoglobin, although early mucosal lesions in CD can be associated with the damage of small capillaries [58], no data on haemorrhagic changes in the blood vessels has been found in literature. An exception to this is angiitis in the outer part of the bowel wall, which manifests through the infiltration of inflammatory cells into vascular adventitia, or formation of granulomas alongside blood vessels. There is no evidence of haemorrhagic lesions [62] containing the products of methaemoglobin. The deposition of melanin (along with lipofuscin) in the intestinal wall is characteristic of colonic melanosis, related with the chronic use of laxatives [63], and there is limited data on its association with CD [64,65]. Given its extremely rare occurrence, the chance of intestinal melanosis occurring in the patients included in the study is unlikely. The inflammatory bowel wall tends to contain crypt abscesses, the contents of which could be considered as both mucoid and protein-rich tissues. However they are occasional, and are only observed in 19% of patients [62]. However, an additional very important consideration influencing the signal intensity of fine and thin structures, such as the bowel wall, is the partial volume effect [66], where the voxel is influenced not only by the properties of the same structure but also by the nature of the adjacent tissues. Typically, achievable DWI resolution is in the order of 2 mm × 2 mm × 2 mm [67]. In our DWIBS protocol, the acquisition voxel size is 2.50 mm (RL) × 2.98 AP (AP) × 6 mm (slice thickness) therefore, within the single voxel, there will be signal contamination from the adjacent media. Since the bowel lumen contains the viscous and proteinaceous chyme, and occasionally fecal admixture, the ADC-DWI_STIR_ values will be influenced not only by suppression of the mesenterial fat tissue, but also by saturation of signal from the bowel content with short T1 relaxation time. Therefore, when measured at a short distance from the intestinal lumen as carried out in the group of adults, ADC-DWI_STIR_ values are artificially lower, compared to ADC values of DWI with selective fat suppression, i.e., DWI_SPIR_.

ADC-DWI_SPIR_ values are lower in paediatric patients than in adults. This is explained by differences in medical history. Although all adult patients had active CD, their medical history prior to the MRE examination was at least six months long (except for one patient whose duration of illness was unknown), whereas all children (except one with an almost 2-year history of CD) were examined no longer than six months after the onset of symptoms. Therefore, the edema component in the pediatric bowel wall was more pronounced, resulting in a greater diffusion restriction when compared to the adult patients. The presence of epithelioid granulomas may further limit diffusion in the inflamed wall.

The explanation of non-difference between ADC-DWI_STIR_ values in adults and children is more complex. In children, there is expressive oedema. The ROI was positioned at the site with the highest signal intensity (within the submucosal layer of the bowel wall), therefore the distance to the intestinal lumen was sufficient to prevent the signal contamination caused by the partial volume effect. Since the history of the disease is longer in the group of adult patients, apart from oedema, fibrosis is also present. In these locations, the bowel wall is thinner, and ADC-DWI STIR values are influenced by the partial volume effect from the bowel content with short T1 time, which artificially lowers the ADC values.

The absence of difference between ADC-DWI_SPIR_ and ADC-DWI_STIR_ values in the children’s group could also be explained with the predominance of the edematous component, which, by increasing the thickness of the intestinal wall, does not allow the partial volume effect to affect the ADC-DWI_STIR_ values measured in the middle of the submucosal layer of the intestinal wall.

The correlation between ADC-DWI_SPIR_ and MaRIA was moderately negative in both adults (r = −0.50, *p* = 0.004) and children (r = −0.54, *p* < 0.0005) being worse than reported by the study group of the Clermont-Ferrand university showing excellent correlation [19,20]. However, in the systematic review and meta-analysis on using diffusion-weighted MRE for evaluating bowel inflammation in CD, Choi et al. states that ADC demonstrates a moderate strength of correlation at best and Clermont score performs better [68]. This is also consistent with our results, since like the studies by the research group from Clermont-Ferrand university, we also observed excellent correlation between DWI-based Clermont score and MaRIA in both adults (rho = 0.93; *p* < 0.0005) and children (rho = 0.98; *p* < 0.0005). However, there might a methodological error in using correlation between MaRIA and Clermont score, as the data to be correlated should be mutually independent, and should not be used if they include more than one observation in any individual [69]. Apart from RCE used in MaRIA and ADC used in the Clermont score, all other three variables (wall thickness, presence of edema and presence of ulcerations), are used in both equations. The use of correlation analysis opposes the conditions in which correlation can be applied, and in the instances of highest probability, could lead to an overestimation of the similarity between MaRIA and Clermont score. The correctness of this statement is supported by the contradiction between correlation of ADC-DWI_STIR_ and the ADC-DWI_STIR_-based Clermont Index with MaRIA, as despite no apparent correlation between ADC-DWI_STIR_ and MaRIA (r = −0.001, *p* = 0.99) in the adult group, and low negative correlation between ADC-DWI_STIR_ and MaRIA in the paediatric group (r = −0.37, *p* = 0.01), the correlation between DWI_STIR_-based Clermont score and MaRIA remained strong in both adults (r = 0.89; *p* < 0.0005) and in children (rho = 0.95; *p* < 0.0005).

For scanning of the study patients, we used DWI_STIR_ and DWI_SPIR_ (DWIBS) sequences included in the MRE protocol, obtained from the repository provided by the manufacturer. DWI_STIR_ sequence differs from DWI_SPIR_ with the inversion recovery pulse applied. The sharpness of contours in DWI_STIR_ images always outperforms the DWI_SPIR_ series, due to better suppression of background signal and less T2 “shine through” effect, provided that all scanning parameters (number of signal averages, voxel size, slice thickness, FOV and DWI directions parameters) are the same. As the voxel size and number of signal averages does not influence the DWI outcome including the ADC value and these sequences are designed to provide the best possible performance, we did not modify or harmonise the scanning parameters used in the protocol.

In our study, we did not use endoscopy as the reference standard but selected the study groups exclusively by visual MRE findings of CD, i.e., thickened, oedematous bowel wall and markedly increased SI in the DWI tracking images of b = 800 s/mm^2^ along with the low SI in the ADC map. Of course, the lack of correlation with the endoscopic picture could be considered to be a limitation of the study. However, the literature provides a broad picture of the correlation not only between MRE and endoscopic findings, but also between MRE and surgery specimens of resected intestinal segments with certain defined criteria, along with the conclusion that MRI is an informative and sufficiently accurate method to assess altered bowel wall. Based on these observations, for several years now, when referring patients for MRE examinations, clinicians do not duplicate their results with the invasive endoscopy that is also cumbersome for patients. Consequently, in 2019, for the first time, the ECCO-ESGAR guidelines came up with a revolutionary statement that radiological cross-sectional imaging methods (and, therefore, MR) can be used as an alternative to endoscopy to assess CD activity [5]. Therefore, although CD had been endoscopically confirmed in all patients included in our study, the results of the MRE examination were not duplicated by the endoscopic and histopathological findings in any cases. It would have been useful to correlate the MR finding with the histopathological picture of the surgical resection specimens. However, surgical resection with subsequent histopathological analysis of the specimen, which would provide the most complete picture of intestinal wall changes, was performed in only one pediatric patient.

In another related research project, our study team assessed the intra-observer agreement of measurements of ADC-DWI_SPIR_ values, ADC-DWI_STIR_ values, and other components of MaRIA and Clermont scores—bowel thickness, presence of ulcers and RCE composed from WSE-preGd and WSI-postGd. The data was recently published. According to our results, there was a systematic difference in the assessment of ulcers, however, no difference was observed between the measurements of the bowel wall thickness, ADC values of DWI with SPIR, ADC values of DWI with STIR as well as measurements performed for assessment of WSI-preGd and WSI post-Gd [70].

In our opinion, the strengths of our research were as follows: (1) the prospective study design, (2) accurate location-by-location comparison in one and the same bowel segment, (3) explicit ROI size not being defined in the previous studies on MaRIA and Clermont score, except for when conducted by the Caruso’s team using the ROI size between 12 and 20 mm^2^. However, our study also faced several limitations: (1) the relatively low number of participants in the study groups, (2) the study population could be subject to selection bias, as both adult and paediatric study groups were not homogenous regarding the duration of the disease. Paediatric patients included in our study had a relatively shorter history than the adult patients; (3) in both post-Gd and DWI images, the ROIs were placed on the site of the maximum SI. After administration of gadolinium contrast media, in some cases the most intense contrast enhancement was predominantly observed in ileal mucosa, however in other cases the enhancement was evenly distributed throughout the intestinal wall. In contrast, in both types of DWI techniques, bowel wall layers were indistinguishable as the diffusion restriction throughout the intestinal walls was equally intense, which could result in differences of positioning ROI between the T1 post-contrast and DWI sequences; (4) the MaRIA studies are based on a comparison of the visual image with the CDEIS—CD endoscopic activity index in adult patients. Unlike adults, estimation of inflammatory activity in children does not rely on endoscopy findings due to its invasiveness, but rather on the Pediatric Crohn’s Disease Activity Index (PCDAI), which correlates poorly with the MaRIA index (r = 0.42, *p* = 0.016) [6]. Its correlation with Clermont score has not yet been estimated, so, in children, the utility of Clermont score is still unclear.

## 5. Conclusions

In Crohn’s disease, DWI MRI sequence with STIR, when compared to DWI MRI sequence with SPIR, is less reliable and is not suitable for quantitative bowel inflammatory activity assessment to be used in Clermont score, in both adult and paediatric patients. DWI MRI sequence with STIR is advisable to provide qualitative visual identification of bowel inflammation foci in adults as well as paediatric patients.

## Figures and Tables

**Figure 1 diagnostics-10-00347-f001:**
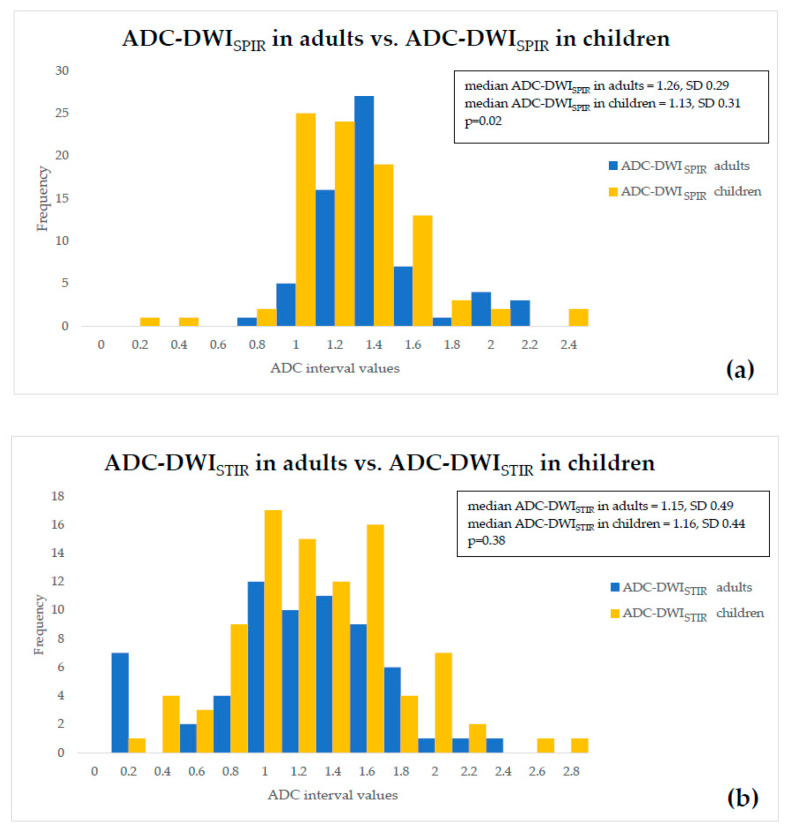
Comparison of (**a**) ADC-DWI_SPIR_ values, and (**b**) ADC-DWI_STIR_ values between adults and children.

**Figure 2 diagnostics-10-00347-f002:**
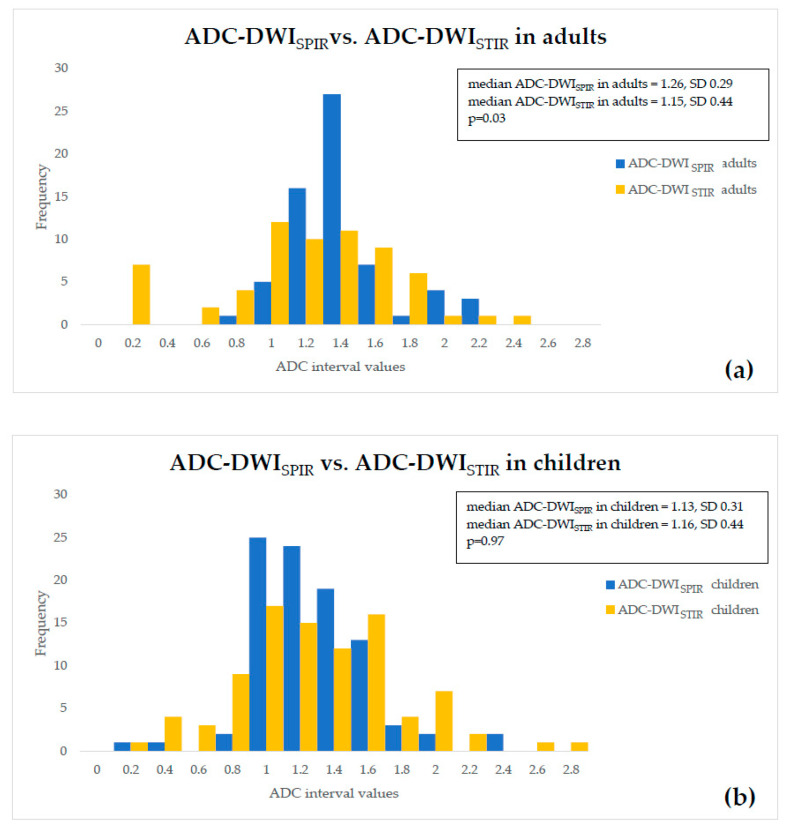
Comparison between ADC-DWI_SPIR_ and ADC-DWI_STIR_ values in the (**a**) adult group and (**b**) in children.

**Figure 3 diagnostics-10-00347-f003:**
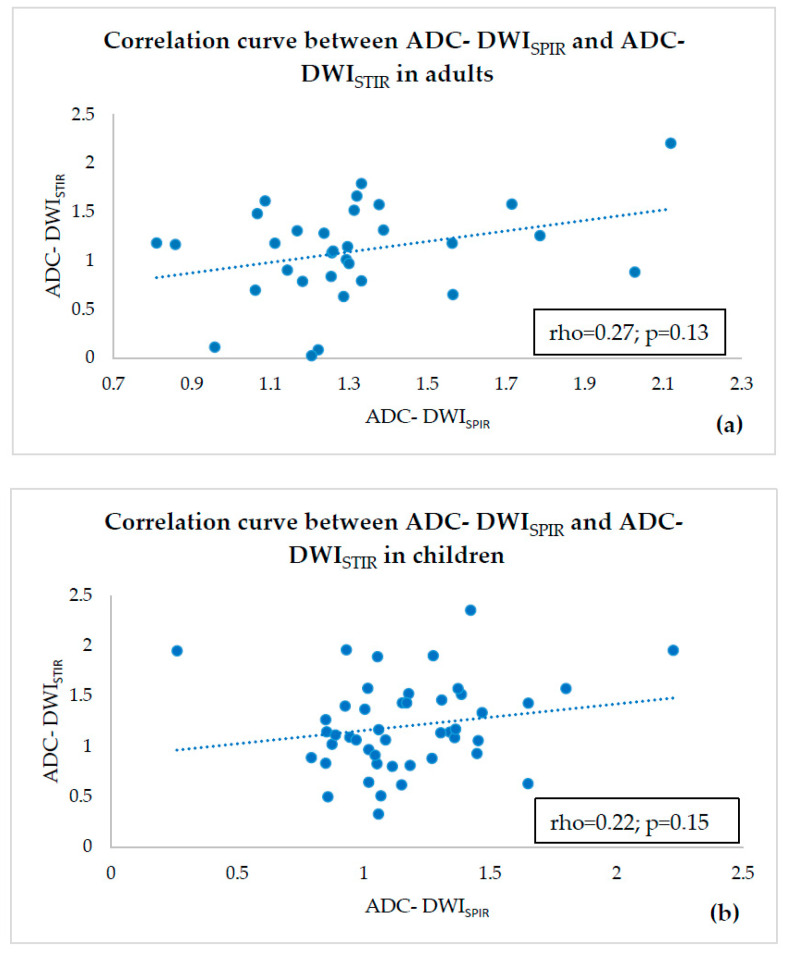
Correlation curve between ADC-DWI_SPIR_ and ADC-DWI_STIR_ in adults (**a**) and children (**b**) showing no correlation.

**Figure 4 diagnostics-10-00347-f004:**
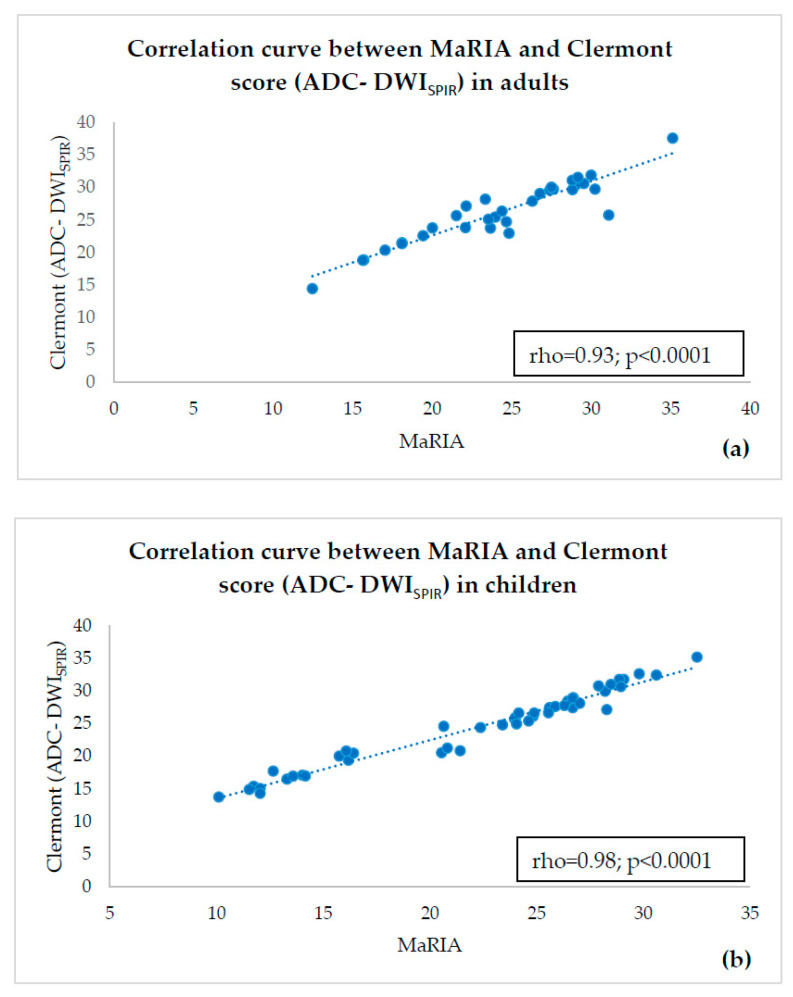
Correlation curve between MaRIA and ADC-DWI_SPIR_—based Clermont score in adults (**a**) and children (**b**) showing high correlation.

**Figure 5 diagnostics-10-00347-f005:**
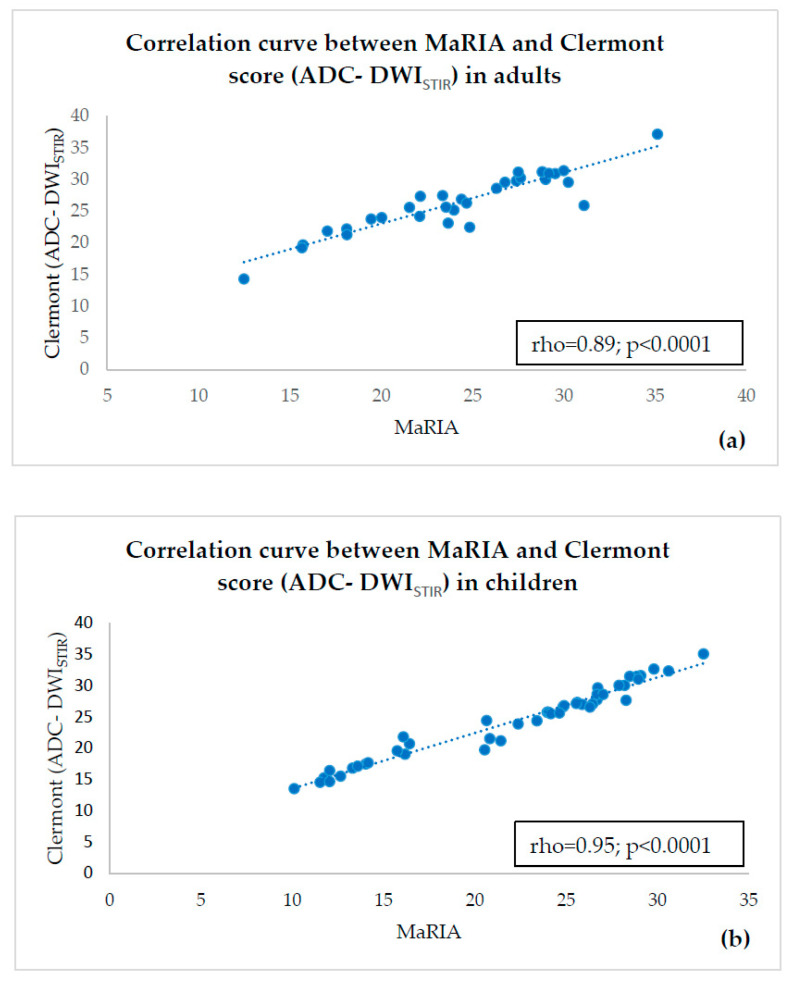
Correlation curve between MaRIA and ADC-DWI_STIR_—based Clermont score in adults (**a**) and children (**b**) showing high correlation.

**Figure 6 diagnostics-10-00347-f006:**
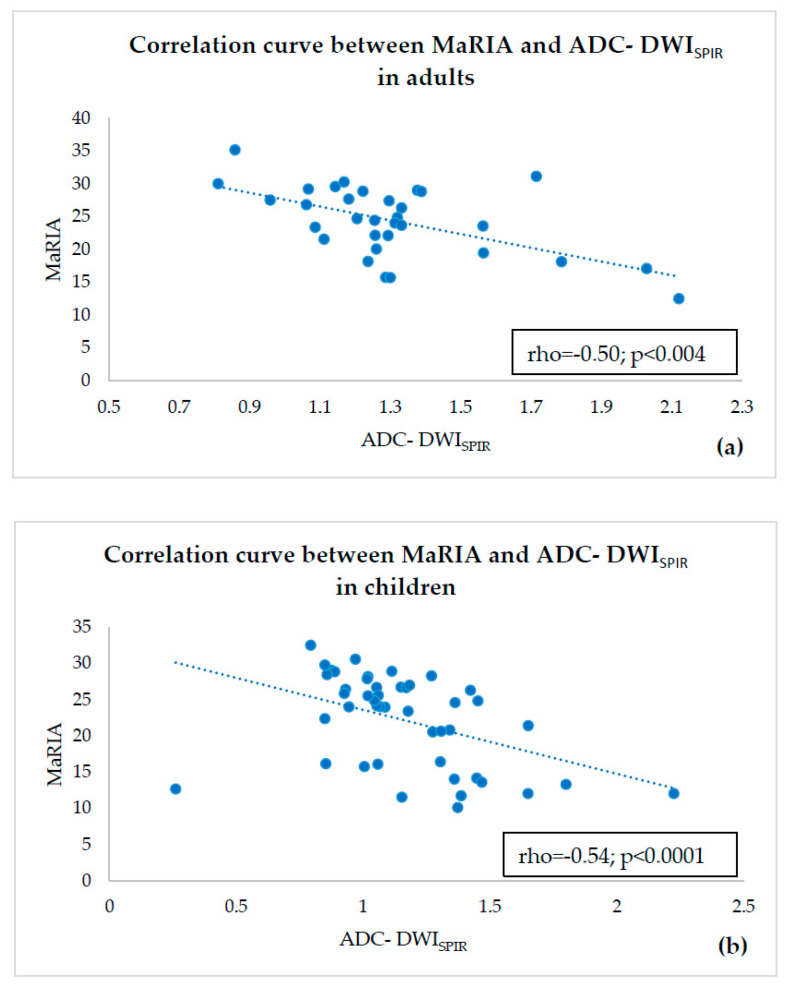
Correlation curve between ADC-DWI_SPIR_ and MaRIA in adults (**a**) and children (**b**) showing moderate negative correlation.

**Figure 7 diagnostics-10-00347-f007:**
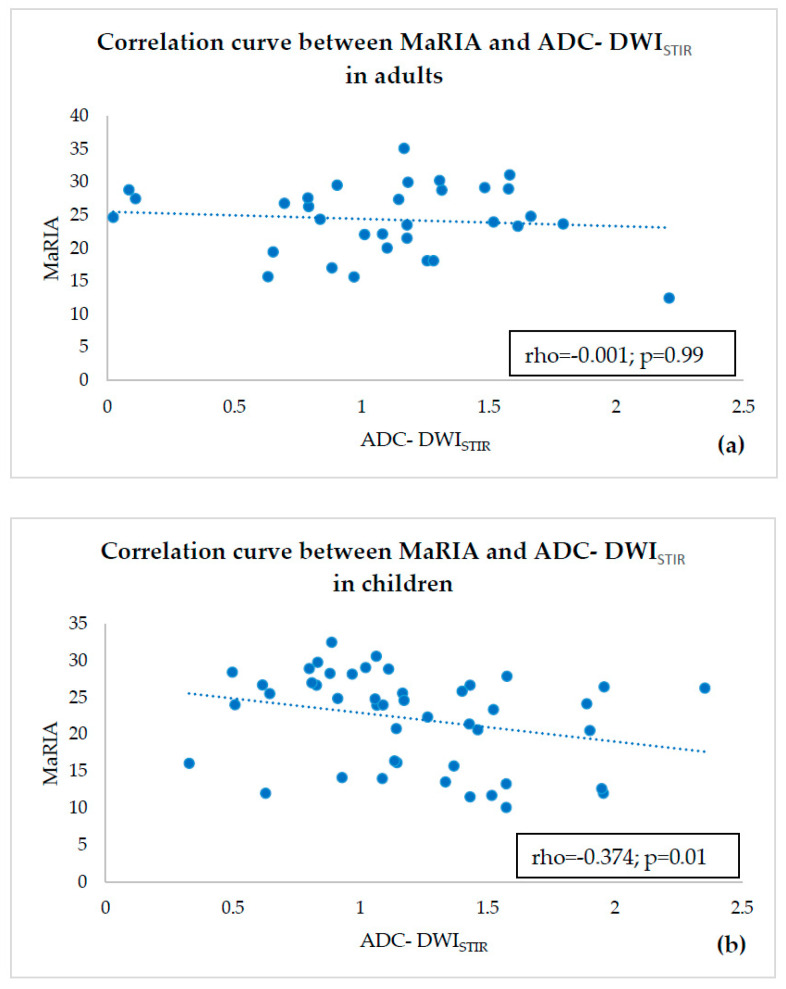
Correlation curve between ADC-DWI_STIR_ and MaRIA in adults (**a**) showing no correlation and in children (**b**) showing low negative correlation.

**Figure 8 diagnostics-10-00347-f008:**
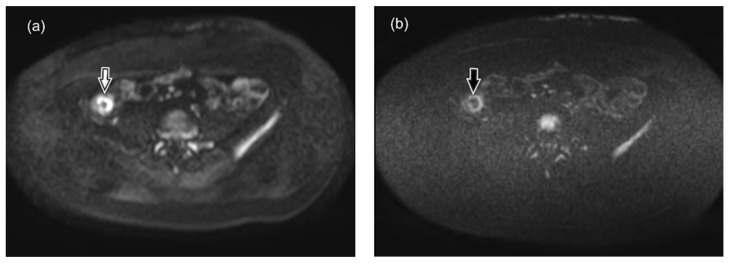
DWI_SPIR_ (**a**) and DWI_STIR_ (**b**) tracking images of b = 800 s/mm^2^ images of 40 years old male patient with active CD. Inflamed bowel walls present high signal intensity. Despite decreased SNR causing graininess in the images, the resolution of inflamed bowel and delineation of contours is better in DWI_STIR_ image (black arrow) as compared to the DWI_SPIR_ image (white arrow).

**Figure 9 diagnostics-10-00347-f009:**
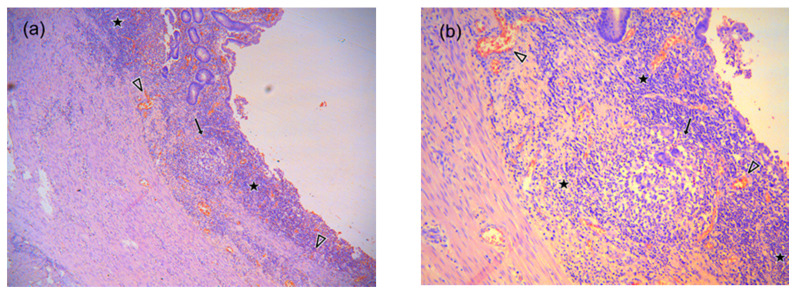
Inflammation of the ileal wall in active chronic CD in 14 y.o. boy, hematoxylin-eosin staining (courtesy of Dr. Ivars Melderis), (**a**) at magnification× 40, (**b**) at magnification× 100. Infiltration by plasma cells, neutrophils and abundant number of lymphocytes is present in the mucosal part of the bowel (star), along with epithelioid granuloma (arrow) in the lamina muscularis mucosae providing additional contribution for restricted diffusion signal in DWI images in children CD. Due to granulation process, there are unaltered red blood cells (arrowheads) in the capillaries of intestinal mucosa. The presence of blood degradation products is not detectable in any of the intestinal wall layers.

**Table 1 diagnostics-10-00347-t001:** Scanning parameters of DWI_SPIR_ and DWI_STIR_ techniques included in the MRE protocol.

Scanning Protocol	DWI_SPIR_ ^1^	DWI_STIR_ (DWIBS)_ ^2^
Sequence	SE-EPI ^3^	STIR-EPI ^4^
Mode	Single shot	Single shot
Coil	SENSE ^5^ body	SENSE body
Slice orientation	Axial	Axial
FOV ^6^	RL ^7^ 400 mm, AP ^8^ 350 mm, FH ^9^ 303 mm	RL 400 mm, AP 350 mm, FH 303 mm
ACQ ^10^ voxel size	RL 3.03 mm × AP 3.57 mm × slice thickness 6 mm	RL 2.50 mm × AP 2.98 mm × slice thickness 6 mm
Reconstruction voxel size	RL 1.79 mm × AP 1.79 mm × slice thickness 6 mm	RL 1.39 mm × AP 1.39 mm × slice thickness 6 mm
Fold-over suppression	No	No
Reconstruction matrix	224	288
SENSE	Yes	Yes
P reduction (AP)	2	2.5
Number of stacks	1	1
Type	Parallel	Parallel
Slices	46	46
Slice gap (mm)	0.6	0.6
Slice orientation	Transverse	Transverse
Fold-over direction	AP	AP
Fat shift direction	A	P
TE ^11^	66 ms	78 ms
TR ^12^	1426 ms	7055 ms
TI ^13^	-	180 ms
Fast imaging mode	EPI ^14^	EPI
Flip angle	90°	
Fat suppression	SPIR	STIR
b factors	0, 600, 800 s/mm^2^	0, 600, 800 s/mm^2^
Respiratory compensation	Trigger	No
Number of signal averages	3	5
Total scan time	4 min. 12 s	5 min. 56 s

^1^ DWI_SPIR_—Diffusion-Weighted imaging with Spectral Presaturation with Inversion Recovery technique. ^2^ DWI_STIR_—Diffusion-Weighted Imaging with Short-Tau Inversion Recovery. ^3^ SE-EPI—Spin Echo—Echo Planar Imaging. ^4^ STIR-EPI—Short T1 Inversion Recovery—Echo Planar Imaging. ^5^ SENSE—SENSitivity Encoding. ^6^ FOV—Field of View. ^7^ RL—Right-Left direction. ^8^ AP—Anterior-Posterior direction.^9^ FH—Foot–Head direction.^10^ ACQ—Acquisition. ^11^ TE—Echo Time. ^12^ TR—Repetition Time. ^13^ TI—Inversion Time. ^14^ EPI—Echo Planar Imaging.

**Table 2 diagnostics-10-00347-t002:** Values of ADC-DWI_SPIR_, ADC-DWI_STIR_, MaRIA as well as ADC-DWI_SPIR_ and ADC-DWI_STIR_ -based Clermont scores in the groups of adult and paediatric patients.

Measurement	N	Minimum Value	Maximum Value	Median Value	SD
ADC-DWI_SPIR_ (mm^2^/s), adults	32	0.66 × 10^−3^	2.16 × 10^−3^	1.26 × 10^−3^	0.29
ADC-DWI_SPIR_ (mm^2^/s), children	46	0.18 × 10^−3^	2.23 × 10^−3^	1.13 × 10^−3^	0.31
ADC-DWI_STIR_ (mm^2^/s), adults	32	0.01 × 10^−3^	2.37 × 10^−3^	1.15 × 10^−3^	0.49
ADC-DWI_STIR_ (mm^2^/s), children	46	0.20 × 10^−3^	2.74 × 10^−3^	1.16 × 10^−3^	0.44
MaRIA, adults	32	10.65	36.65	24.43	5.31
MaRIA, children	46	9.96	37.67	22.08	6.67
DWI_SPIR_-based Clermont score, adults	32	12.85	39.23	26.23	4.76
DWI_SPIR_-based Clermont score, children	46	13.59	40.74	23.53	5.42
DWI_STIR_-based Clermont score, adults	32	5.92	38.78	24.28	4.65
DWI_STIR_-based Clermont score, children	46	8.25	39.52	24.39	5.77

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
