# Peer review of "Comparison between Diffusion-Weighted Sequences with Selective and Non-Selective Fat Suppression in the Evaluation of Crohn’s Disease Activity: Are They Equally Useful?"

_diagnostics, 2020, doi:10.3390/diagnostics10060347_

Round 1
Reviewer 1 Report
This paper covers an important technique for the evaluation of Crohn's disease. As we move towards minimally invasive techniques for the assessment of patients with this disease, we must be sure of the effectiveness and reliability of the new approaches. This paper examines the differences between two diffusion-weighted MR imaging sequences and describes a study to obtain quantitative differences.
I found this paper to be clear, interesting, well-written and well-structured, describing a well-designed study. In particular, the introduction gives excellent background and a useful set of references is provided. The results are presented very clearly. Inevitably, it would have been good to enrol more patients into the study (but the paper mentions this and takes account of the consequences). I have a few *very* minor corrections.
line 43 - visualization of bowels
line 50 - add full stop
line 73 - several pixels away
line 115 - to therapy)
line 200- maybe put in a reference for the MaRIA formula?
line 238 - in Table 2.
lines 294 and 344 - use "best" instead of "most optimal".
line 355 - remove the space after SPIR-
line 446 - in the order
line 484 - we did not use endoscopy
There are also a few minor capitalisation errors in the references, but nothing significant.
Nice piece of work - well-written paper.
Author Response
Dear Sir/Madam,
Thank you for the the job you have done reviewing our publication, and for your awareness pointing out errors and inaccuracies.
According to your suggestion, we perform the following corrections:
Line 43 - visualization of bowels: changes done
Line 50 - add full stop: changes done
Line 73 - several pixels away: changes done
Line 115 - to therapy): changes done
Line 200 - maybe put in a reference for the MaRIA formula?:
Changes are not needed here since the reference is given in the line 206. In the lines 200-206, calculation of MaRIA is described, along with calculation of RCE what is a component of MaRIA, and all this information comes from one article that is referenced as (39) in line 206.
Line 238 - in Table 2: changes are done
Line 294 and 344 - use "best" instead of "most optimal": changes are done
Line 355 - remove the space after SPIR-:
in this sentence, we changed from “navigator triggered SPIR-, SPAIR-, and CHESS-based DWI techniques” to “DWI sequences with SPIR, SPAIR, and CHESS”. In the revised version, removal of space is not required.
Line 446 - in the order: changes done
Line 484 - we did not use endoscopy: changes done
There are also a few minor capitalisation errors in the references, but nothing significant: According to your remark, we reviewed the references and corrected the capitalization in some of them.
While waiting for a revision from the journal, we also decided that several improvements have to be made in the article:
- The first author added the second affiliation because of being in both affiliations.
- We replaced the form “STIR-, SPIR-, SPAIR-, or CHESS-based DWI sequences with DWI with SPIR, DWI with STIR, DWI with SPAIR, or DWI with CHESS since the fat suppresion technique is combined with DWI but not is not a physical basis of it.
- We decided to briefly outline that the DWI-SPIR and DWIBS protocols used were taken from the protocol repository provided by the manufactured and were not mutually matched, acknowledging that this does not significantly affect the ADC measurement results.
- Since during the time of the revision our article on intra-observer agreement in measurements forming MaRIA and Clermont scores has been published, we upgrade the text along with inserting the reference.
Reviewer 2 Report
Apine and colleagues presented a very innovative study aimed to establish the diagnostic accuracy of Diffusion-Weighted Imaging (DWI) with selective and non-selective fat suppression comparing the results of such techniques with the current adopted MaRIA score. Overall, the manuscript is well written and the experimental design well-conceived. Below are reported only few minor revisions that will improve the quality of the manuscript:
1) It is suggested to shorten both Introduction and Discussion sections in order to make the manuscript more readable;
2) In line 46, the authors should use the term Inflammatory Bowel Disease and then the acronym IBD;
3) Have the authors consider to perform additional statistical analyses to establish the accuracy of DWIspir and DWIstir in CD patients? Please, if possible, consider to perform ROC and Cox regression;
4) In the Introduction or Discussion section, the authors should briefly discuss the use of these novel imaging techniques for the early diagnosis of Chron’s disease. It is well established that early diagnosis of CD is associated with a reduction of intestinal surgery interventions. Please discuss this important issue clarifying if DWIspir and DWIstir may be used for this purpose;
5) Could the ADC-DWISTIR and ADC-DWIspir values be lower in children than in adults because of the shorter CD history? Please discuss better this issue;
6) In line 446 should “mm3” be “mm”? Please check.
Author Response
Dear Reviewer,
Thank you for the the job you have done reviewing our publication and for your very valuable advices regarding corrections to be done. Hereby, please, find corrections performed according to your suggestions.
- It is suggested to shorten both Introduction and Discussion sections in order to make the manuscript more readable:
We did shorten both Introduction and Discussion sections making it better readable.
Introduction:
The 1st paragraph (Line 35-50): text shortened and made more compact.
The 3rd paragraph (Line 60-68): text shortened. Also, according to your suggestion to discuss early importance of CD diagnosis, a sentence added regarding improved sensitivity of MRE in DC diagnostics, when added DWI sequence. To describe DWI more clearly, a part on ADC measurements and their importance in the Clermont score, is removed and put in the next paragraph.
The 5th paragraph (Line 69-75) shortened and made more compact.
Line 76-122: the paragraph splitted into two paragraphs separating description of fat selective DWI from fat non-selective DWI. A sentence is added resuming the most important idea about fat selective DWI.
Discussion:
1st paragraph (Line 293-308) shortened explaining the importance of assessment of transmural healing in a more compact way. A sentence on importance of using fat suppression technique with DWI is also removed since this is outlined in the Intraduction.
2nd paragraph (Line 309-326) also shortened chosing words explaining idea in more compact way.
Lines 384-390: two paragraphs combined into one.
Lines 388-390: The reasons for the ADC differences between DWI SPIR and DWI STIR have been redefined.
Line 391-410: characterization of histhopatological changes in Crohn’s diseases leading to restricted diffusion made more compact and significantly shortened. The sentences “Unfortunately, we did not have the opportunity to perform a comprehensive correlation of the altered bowel structure with the MRI findings, since surgical resection and comprehensive evaluation of the altered bowel segment with MRI data was possible in only one pediatric patient. The remaining pediatric patients and all adults were subjected to endoscopic examination only, which did not provide information on the transmural intestinal wall.” removed, and a note of this idea “Surgical resection with subsequent histopathological analysis of the specimen which would give the most complete picture of intestinal wall changes, was performed in only one pediatric patient.” is added in the paragraph that starts with Line 643 (in the revised version)
Lines 419-462: Two paragraphs combined into one. Much of the text has been reworded in clearer wording.
Line 473: "The equations for calculation of MaRIA and Clermont score are not linearly related” removed as redundant.
Paragraph 499-505: changes made. During the time of review of this article, our article about intraobserver agreement was published. Therefore, the text of the paragraph is corrected accordingly, and reference to the published article is added.
2. In line 46, the authors should use the term Inflammatory Bowel Disease and then the acronym IBD:
The acronym IBD changed to the full term “Inflammatory Bowel Disease”.
3. Have the authors consider to perform additional statistical analyses to establish the accuracy of DWIspir and DWIstir in CD patients? Please, if possible, consider to perform ROC and Cox regression;
At the current stage of the research project, we cannot do much about further analysis of DWIspir and DWIstir to determine the accuracy of Crohn's disease patients. As mentioned in the Discussion section, we did not use neither endoscopy nor surgical specimens as the reference standard, but mutually compared both of the sequences and correlated the ADC values of both sequences with the MaRIA index.The most accurate estimation would be comparing the corresponding ADC values with morphohistological patterns of surgical specimens. In this research, surgical resection was performed in only one patient that is insufficient for accuracy estimations and for building a ROC curve. ROC analysis will be possible later, when we collect sample size of resection specimens enough large to calculate diagnostic accuracy of ADC of both DWI sequences.
In our study, we analysed unique patients. We did not use Cox regression since follow-up analysis what was out of the scope of our research.
4. In the Introduction or Discussion section, the authors should briefly discuss the use of these novel imaging techniques for the early diagnosis of Chron’s disease. It is well established that early diagnosis of CD is associated with a reduction of intestinal surgery interventions. Please discuss this important issue clarifying if DWIspir and DWIstir may be used for this purpose;
Diagnosing CD and evaluation its activity are two separate directions of using DWI in CD. The performance of DWI in early diagnostics of CD was not discussed in details, since the scope of our study was evaluation of CD activity in already diagnosed CD. However, according to your suggestions, we supplied the Introduction section with a sentence that, when added to the MRE protocol, DWI yields better sensitivity in diagnostics of CD than conventional MRE alone, thus potentially favoring early diagnosis of CD. Also, importance of precise delineation of inflammatory lesions to avoid delayed diagnosis is described in the Discussion section.
5. Could the ADC-DWISTIR and ADC-DWIspir values be lower in children than in adults because of the shorter CD history? Please discuss better this issue;
According to our research, children had lower ADC-DWISPIR values than adults. The shorter CD history could be the reason because the edema component was more pronounced. Also, iun children, there are more granulomas that add to restriction of diffusion resulting in lower ADC values comparing to adults.
Regarding ADC-DWISTIR values, there is no difference between children and adults. In children, oedema is so expressive that, when ROI is put on the middle layer of the bowel wall, there is signal influence from the bowel content, due to partial volume effect. Since in adults history of the disease is longer, apart from oedema, fibrosis is present therefore the bowel is thinner, and therefore, ADC-DWI STIR values are influenced by the partial volume effect from the bowel content with short T1 time which artificially lowers the ADC value.
Both of these statements are implemented in the revised Discussion section.
6. In line 446 should “mm3” be “mm”? Please check.
Yes, you are right. However, this form of the text was taken from the reference: B. Scherrer, A. Gholipour, and S. K. Warfield, “Super-Resolution in Diffusion-Weighted Imaging Benoit,” Med Image Comput Comput Assist Interv., vol. 14, no. Pt2, pp. 124–132, 2011. In the text they state exactly “typically achievable DWI resolution is on the order of 2×2×2mm3”. Since this is not quite correct, we change it to “achievable DWI resolution is in the order of 2 mm × 2 mm × 2 mm”.
While waiting for a revision from the journal, we also decided that several improvements have to be made in the article:
- The first author added the second affiliation because of being in both affiliations.
- We replaced the form “STIR-, SPIR-, SPAIR-, or CHESS-based DWI sequences with DWI with SPIR, DWI with STIR, DWI with SPAIR, or DWI with CHESS since the fat suppresion technique is combined with DWI but not is not a physical basis of it.
- We decided to briefly outline that the DWI-SPIR and DWIBS protocols used were taken from the protocol repository provided by the manufactured and were not mutually matched, acknowledging that this does not significantly affect the ADC measurement results.
- Since during the time of the revision our article on intra-observer agreement in measurements forming MaRIA and Clermont scores has been published, we upgraded the text along with inserting the reference.